# A Nomogram Based on Atelectasis/Obstructive Pneumonitis Could Predict the Metastasis of Lymph Nodes and Postoperative Survival of Pathological N0 Classification in Non-small Cell Lung Cancer Patients

**DOI:** 10.3390/biomedicines11020333

**Published:** 2023-01-24

**Authors:** Yi-Han Liu, Lei-Lei Wu, Jia-Yi Qian, Zhi-Xin Li, Min-Xing Shi, Zi-Ran Wang, Long-Yan Xie, Yu’e Liu, Dong Xie, Wei-Jun Cao

**Affiliations:** 1Department of Respiratory and Critical Care Medicine, Shanghai Pulmonary Hospital, Institute of Respiratory Medicine, School of Medicine, Tongji University, Shanghai 200433, China; 2Department of Thoracic Surgery, Shanghai Pulmonary Hospital, School of Medicine, Tongji University, Shanghai 200433, China; 3Department of Orthopedics, The 903rd Hospital of PLA, Hangzhou 310007, China; 4School of Medicine, Tongji University, Shanghai 200092, China

**Keywords:** atelectasis, lymph node metastasis, NSCLC, nomogram, obstructive pneumonitis

## Abstract

The eighth TNM staging system proposal classifies lung cancer with partial or complete atelectasis/obstructive pneumonia into the T2 category. We aimed to develop nomograms to predict the possibility of lymph node metastasis (LNM) and the prognosis for NSCLC based on atelectasis and obstructive pneumonitis. Methods: NSCLC patients over 20 years old diagnosed between 2004 and 2015 were selected from the Surveillance, Epidemiology, and End Results (SEER) database. The nomograms were based on risk factors that were identified by Logistic regression. The area under the receiver operating characteristic (ROC) curve (AUC) was performed to confirm the predictive values of our nomograms. Cox proportional hazards analysis and Kaplan–Meier survival analysis were also used in this study. Results: A total of 470,283 patients were enrolled. Atelectasis/obstructive pneumonitis, age, gender, race, histologic types, grade, and tumor size were defined as independent predictive factors; then, these seven factors were integrated to establish nomograms of LNM. The AUC is 0.70 (95% CI: 0.694–0.704). Moreover, the Cox proportional hazards analysis and Kaplan–Meier survival analysis showed that the scores derived from the nomograms were significantly correlated with the survival of pathological N0 classification. Conclusion: Nomograms based on atelectasis/obstructive pneumonitis were developed and validated to predict LNM and the postoperative prognosis of NSCLC.

## 1. Introduction

Lung cancer is the most common cause of cancer-related deaths worldwide, causing over 1.8 million deaths annually [1]. There were about 235,760 new cases diagnosed with lung cancer in the US in 2021. Non-small cell lung cancer (NSCLC) is the most common histological type of lung cancer, occupying around 85% of lung cancers. NSCLC mainly includes lung squamous cell carcinoma (SCC) and lung adenocarcinoma (ADC) [2]. Treatments for lung cancer include surgery, chemotherapy, radiotherapy, and targeted therapy. The prognosis for lung cancer patients remains unsatisfactory, especially those with metastases [3]. Lymph node metastasis (LNM) is a poor prognosticator in NSCLC [4]. Therefore, early detection of LNM in early-stage NSCLC is crucial for patients’ risk stratification and therapeutic planning in clinical practice.

Atelectasis and obstructive pneumonitis are common complications in lung cancer patients, mostly discovered during initial diagnosis. They are caused by partial or complete obstruction of a tracheal bronchus by cancer tissue, and they will lead to lung tissue shrinking or recurrent infection in the same position. Tumor-associated atelectasis and obstructive pneumonitis can cause cough, chest pain, chest tightness, hemoptysis, dyspnea, and even death and is an essential factor affecting survival. In previous studies, the role of atelectasis and obstructive pneumonitis on cancer prognosis has remained controversial. Some studies demonstrated that atelectasis was considered a poor prognostic sign and confirmed as one of the (tumor) T classification factors in the staging systems [5,6]. There are a number of studies showing the benefits of adaptive radiotherapy in NSCLC patients with atelectasis/obstructive pneumonia [7,8]. Still, few discuss whether atelectasis/obstructive pneumonia help physicians judge whether a patient requires neoadjuvant chemotherapy. Our study is meant to explore the potential relationship between atelectasis/obstructive pneumonia and lymph node metastasis in patients, and to provide recommendations for clinicians evaluating the lymph node situation and developing treatment regimens for patients with NSCLC complicated by atelectasis/obstructive pneumonia.

Currently, there is only limited information on estimating the risk factors related to LNM in NSCLC individuals. This study performed multivariate Logistic regression screening for atelectasis/obstructive pneumonitis and other clinical variables with predictive value. Then, we established nomograms based on atelectasis/obstructive pneumonitis to confer risk prediction of LNM and the postoperative prognosis in NSCLC patients.

## 2. Patients and Methods

### 2.1. Patients

We retrieved all the clinical information about 44,203 NSCLC cases from the Surveillance, Epidemiology, and End Results (SEER) database. The tumor, nodes, and metastasis (TNM) staging system was determined according to the classification of the 8th American Joint Committee on Cancer (AJCC) [9] Staging Manual. In this study, all patients we recruited met the following criteria: (1) patients were diagnosed with NSCLC between 2004 and 2015; (2) patients ≥20 years old and were active in follow-up. The criteria of exclusion were as follows: (1) patients were diagnosed with other malignant diseases; (2) patients whose T, N, and M classifications were unclear; (3) patients diagnosed as N3 or/ M1 classifications; and (4) patients who died within one month of diagnosis. The flow diagram is presented in Figure 1.

### 2.2. Statistical Methods

R version 3.6.1 (R Fundamental for Data Science, Vienna, Austria) (https://www.r-project.org/, accessed on 1 December 2021) and SPSS 23.0 (IBM Corp., Armonk, NY, USA) were applied to do all the statistical analysis. Categorical variables were described by counts and percentages. Pearson’s chi-square test was used for the comparison of categorical variables. We performed two multivariate Logistic regression analyses to evaluate the influence of atelectasis/obstructive pneumonitis, gender, race, grade, tumor size, histological, and age on LNM. During our multivariable Logistic regression analyses, risk ratios (RRs) and 95% confidence intervals (CIs) were calculated. Statistical significance was considered when a two-tailed *p*-value < 0.05.

Two nomograms [10,11] were constructed based on seven factors: atelectasis/obstructive pneumonitis, age, gender, race, histology, grade, and tumor size. These two nomograms were used to predict the possibility of N1/N2 and N2, respectively. The predicted power of these nomograms was assessed by the receiver operating characteristic (ROC) curve, the higher area the under the curve (AUC) was corresponded to the better accuracy.

The patients were divided into low or high score groups based on nomograms; then, the survival curves were drawn by the Kaplan–Meier (KM) survival analysis and compared by the Log-rank test. Finally, to evaluate the influence of scores calculated from nomogram, radiotherapy, chemotherapy, surgical type, and lymph nodes on cancer-specific mortality in patients with N0 classification after surgery, multivariate Cox regression analysis was performed to calculate hazard ratio (HR) and 95% CI for cancer-specific mortality.

## 3. Results

### 3.1. Patient Characteristics

Overall, we identified 44,203 patients from the SEER database. The gender distribution was approximately the same (50.5% were male, 49.5% were female). About 67.9% of these patients were over the age of 65 years. The upper location of the tumor in the lung had a distribution of 59.6%. About 42.3% of these patients selected lobectomy, while 45.9% of patients had conservative treatment. ADC was the most common histological type (48.3%). The clinical information of patients in this study is shown in Table 1. We also found that gender, age, location, surgical type, receiving radiotherapy or chemotherapy, tumor size, N classification, and histological type significantly influenced the occurrence of atelectasis and obstructive pneumonitis.

### 3.2. Multivariate Analyses

The results of two multivariate Logistic regression analyses to predict lymph node metastasis are presented in Table 2. After adjustment for other factors, we found that atelectasis/obstructive pneumonitis (yes vs. no, *p* < 0.001), male sex (vs. female, *p* < 0.001), race (other vs. Caucasians, *p* = 0.001), and younger age (<65 vs. >64 years, *p* < 0.001) appeared as risk factors of lymph node metastasis. There are other factors, grade (*p* < 0.001), tumor size (*p* < 0.001), and histological type (*p* < 0.001), that had an impact on lymph node metastasis. The factors mentioned above were established as independent factors and then used to construct the nomograms.

### 3.3. Establishment of the Nomogram

According to the results from the multivariate Logistic regression analyses, atelectasis/obstructive pneumonitis, age, gender, race, histologic types, grade, and tumor size were defined as independent prognostic factors. These seven factors were integrated to establish nomograms of N1/N2 or N2 (Figure 2 and Figure 3).

### 3.4. Verification of the Nomogram and Survival Analysis

We also used the ROC curve to verify the actual predictive power of the nomogram performance of N1/N2 (Figure 4). Moreover, the ROC curves illustrated the prediction model preferable by the AUC value of 0.70, which illustrated that this nomogram model had good accuracy in predicting lymph node metastasis. According to the first nomogram prediction model, we calculated the score of each sample, then divided each sample into high score groups or low score groups by the median. Survival curves revealed that the patients with high scores had poorer mortality than patients with a low score (unadjusted HR = 1.458, 95% CI 1.363–1.559, *p* < 0.001, Figure 4b).

A multivariate Cox regression analysis to evaluate the cancer-specific mortality in patients with N0 classification after surgery (Table 3) and found that higher scores (*p* < 0.001), received radiotherapy (*p* < 0.001), and chemotherapy (*p* < 0.001) appeared as risk factors for cancer-specific mortality. Moreover, surgical types, lymph nodes, and even marital status could significantly impact cancer-specific mortality.

To confirm the prognostic impact of atelectasis/obstructive pneumonitis on the cancer-specific survival of NSCLC patients, we conducted the Kaplan–Meier analysis and the univariable analysis. The results showed that the patients with atelectasis/obstructive pneumonitis had poorer survival than patients without them (HR = 2.053, 95% CI 1.965–2.145, *p* < 0.001, Figure 5).

## 4. Discussion

Lung cancer is the most common malignancy globally, leading the world in both morbidity and mortality. NSCLC is a primary histology type, which constitutes 85% of all lung cancer cases. LNM is a concern for clinicians who treat NSCLC patients due to its effects on patients’ prognosis and guiding therapeutic strategies [12,13]. However, how to make an accurate judgment on LNM before the next treatment remains a challenge to clinicians. In daily clinical practice, some thoracic surgeons perform lymph node biopsy or dissection on the basis of experience. Lymph node dissection can help to remove lymph nodes and clarify the stage of NSCLC, guiding postoperative treatment planning for patients. However, lymph node dissection prolongs the operation time and brings additional surgical risks [14]. PET-CT, lymph node biopsy, mediastinoscopy, and surgery are standard examinations used to detect LNM. The gold standard for diagnosing LNM in NSCLC is a pathological biopsy. PET-CT is a simpler examination method that can evaluate LNM initially, but the price is higher and could be falsely negative or falsely positive [15]. Therefore, more effective and non-invasive methods to evaluate LNM in NSCLC are urgently needed. Nomograms are reliable tools to quantify risk and enhance outcome prediction, which has been widely accepted [16,17]. According to these predictive models, the selections of examination and treatment could be more reasonable. A small number of studies have been published to date that predict LNM in patients with NSCLC [18,19]. Our study focused on predicting the occurrence of LNM in NSCLC patients with atelectasis/obstructive pneumonitis, in order to provide a more accurate prediction in such patients.

In lung cancer, atelectasis and obstructive pneumonitis commonly happen due to endobronchial obstruction. About one-quarter to one-third of patients with lung cancer are found to have atelectasis at first presentation [20,21]. Atelectasis and obstructive pneumonitis are important prognostic factors that can cause cough, pleuritic chest pain, dyspnea, and even death. Ou et al. confirmed that hilar atelectasis or obstructive pneumonitis were related to poor prognosis in NSCLC patients with tumor size >3 cm [22]. However, the study of Coen et al. showed that atelectasis had no influence on outcomes in NSCLC patients [23]. Dediu and Bulbul et al. believed that the positive prognostic value of atelectasis/obstructive pneumonitis in patients with lung cancer might be due to the specific growth pattern and decreased intratumoral blood flow [24,25]. Moller and Bin Wang thought patients with atelectasis/obstructive pneumonitis might benefit more from adaptive radiotherapy [7,8]. Thus far, whether there is an effect of atelectasis/obstructive pneumonitis on the benefit of chemoradiotherapy and the overall survival of NSCLC patients has not been fully elucidated.

There are two main problems in the clinical evaluation of NSCLC patients with atelectasis/obstructive pneumonia. One is that when lung cancer is complicated with atelectasis/obstructive pneumonia, it is difficult to determine the specific area of the tumor through CT and other imaging. Second, it is difficult to decide on the spread of tumors in lung tissue, especially when mediastinal lymph nodes overlap with atelectasis. Therefore, for NSCLC patients with atelectasis/obstructive pneumonia, there is an urgent need for a good model to help clinicians better evaluate the patient’s condition and formulate treatment plans. In this present study, we aimed to figure out the controversial issue and estimate the potential impact of atelectasis and obstructive pneumonitis on LNM and cancer-specific mortality in NSCLC. After finding out these important clinical variables, including atelectasis and obstructive pneumonitis, which are related to the N stage through multivariate Logistic regression analyses, the predictive nomograms for the N1-2 stage or N2 stage were constructed. Based on the risk score calculated by the multivariate Logistic regression for the N stage, KM survival analysis and a multivariate Cox regression analysis to evaluate the cancer-specific mortality in patients with N0 classification were performed. The final results suggest that atelectasis and obstructive pneumonia are associated with LNM and can negatively predict the prognosis of patients with NSCLC.

Concretely, we constructed a nomogram to prognose LNM with good predictive power (AUC = 0.70). The predictive nomograms for LNM included seven clinical variables, atelectasis/obstructive pneumonitis, histology, pathological differentiation, tumor size, age, and gender, as well as race, which were roughly consistent with the results in the previous studies [18,26,27,28,29]. Moreover, the abovementioned seven factors included in the nomograms could be obtained before surgery. Therefore, the nomogram could provide reference information to doctors to decide on a neoadjuvant treatment and surgical plan. According to our analysis, NSCLC patients with atelectasis/obstructive pneumonitis, poorly differentiated or undifferentiated grade, and tumor size of more than 3.0 cm were more inclined to have LNM. Cox proportional hazards analysis and Kaplan–Meier survival analysis proved that the high score was significantly correlated with patient postoperative survival of N0 classification in NSCLC, hazard ratio (HR) = 1.458 (95% CI: 1.363–1.559). Therefore, we suggest clinical physicians take these patients with high scores more seriously than others.

Our report had considerable strengths. Firstly, we established the first nomograms, which were based on atelectasis/obstructive pneumonitis, and estimated the clinical risk factors associated with LNM in NSCLC patients. Secondly, a large number of cases with NSCLC were extracted from the SEER database, which made our conclusion more convincing. There are several inevitable limitations that should be acknowledged in this study. First, the parameters analyzed in this study are not comprehensive because the quantity of data released in SEER is limited; thus, potential bias or error could be caused. Second, hematological indicators and tumor markers are not included in our study. Third, all data series downloaded for performing the prediction nomograms came from Western countries, so we should be cautious when applying the conclusion of our study to NSCLC patients from Asian countries. Fourth, according to the SEER database, we only included traditional surgery, radiotherapy, and chemotherapy, and did not add other treatments such as neoadjuvant therapy to predict and evaluate LNM. Fifth, the study was retrospective. Therefore, selection bias was inevitable. For example, the distribution of marital status was unbalanced in the grouping of atelectasis/obstructive pneumonitis. Based on the above deficiencies, we need to collect enough data to verify the significance of atelectasis in the overall evaluation and treatment decision of NSCLC patients.

## 5. Conclusions

In our study, two nomograms were constructed to predict the LNM of NSCLC patients based on atelectasis/obstructive pneumonitis, and the high AUC values could demonstrate the utility properly. The established prediction nomogram might provide some needed comprehensive clinical dates for improving the personalized LNM prediction of NSCLC patients. Moreover, our study inferred that atelectasis/obstructive pneumonitis causes a negative effect on the postoperative cancer-specific mortality of patients with N0 classification.

## Figures and Tables

**Figure 1 biomedicines-11-00333-f001:**
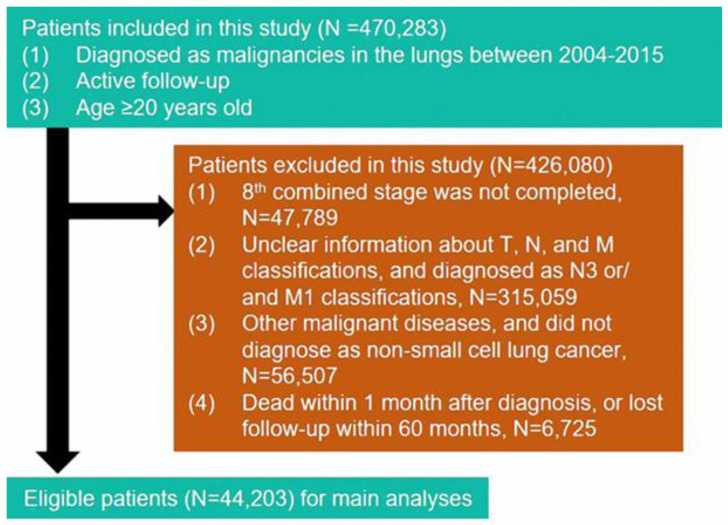
Flow chart of patient screening.

**Figure 2 biomedicines-11-00333-f002:**
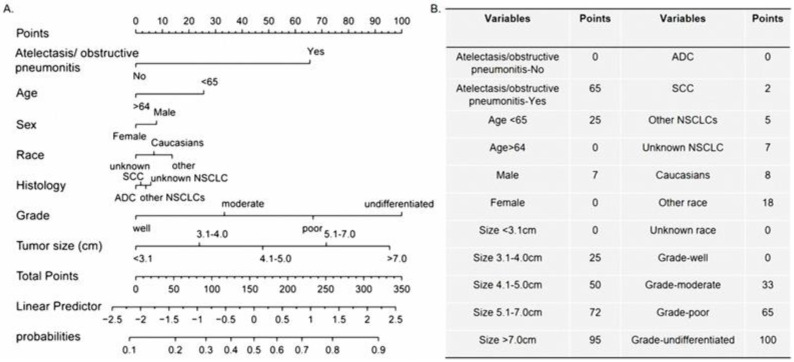
Nomogram predicting lymph node metastasis (N1/N2 stage) for non-small cell lung cancer. (**A**). Scoring rubric of the nomogram (**B**).

**Figure 3 biomedicines-11-00333-f003:**
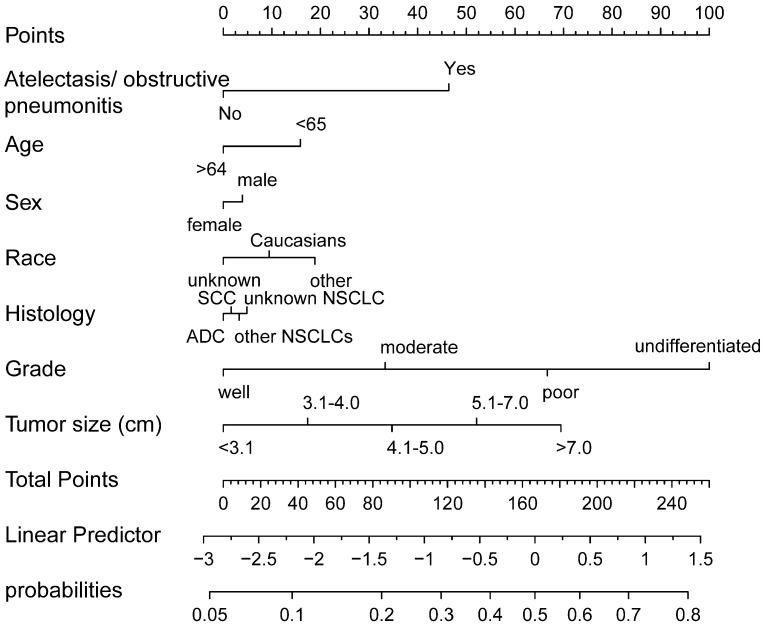
Nomogram predicting N2 stage for non-small cell lung cancer.

**Figure 4 biomedicines-11-00333-f004:**
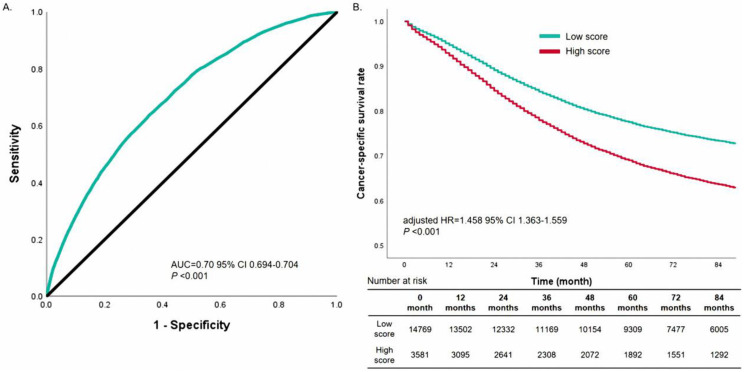
Receiver operating characteristic curve analysis for the sensitivity and specificity of the nomogram system to predict lymph node metastasis (N1/N2 stage) (**A**). Stratified effect of the nomogram (**B**).

**Figure 5 biomedicines-11-00333-f005:**
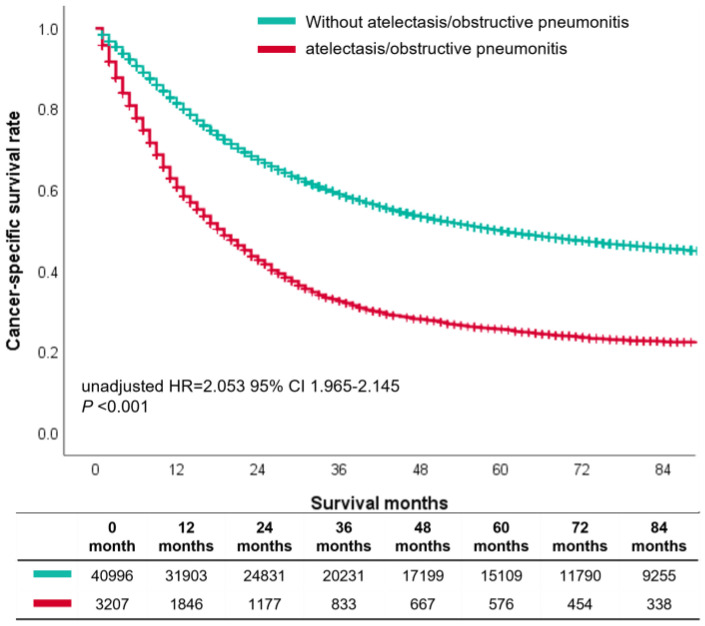
Stratified effect of the atelectasis/obstructive pneumonitis.

**Table 1 biomedicines-11-00333-t001:** Patient characteristics in this study.

Characteristics		All	Atelectasis/Obstructive Pneumonitis		*p*-Value
		*N* = 44,203	No (*N* = 40,996)	Yes (*N* = 3207)	
		Number of Patients (%)			
**Sex**	Male	22,321 (50.5)	20,442 (49.9)	1879 (58.6)	<0.001
	Female	21,882 (49.5)	20,554 (50.1)	1328 (41.4)	
**Race**	Caucasians	36,185 (81.9)	33,598 (82.0)	2587 (80.7)	0.110 *
	Other	7963 (18.0)	7345 (17.9)	618 (19.2)	
	Unknown	55 (0.1)	53 (0.1)	2 (0.1)	
**Year at diagnosis**	2004–2007	14,549 (32.9)	13,368 (32.6)	1181 (36.8)	<0.001
	2008–2011	16,509 (37.3)	15,465 (37.7)	1044 (32.6)	
	2012–2015	13,145 (29.8)	12,163 (29.7)	982 (30.6)	
**Location**	Upper	26,360 (59.6)	24,686 (60.2)	1674 (52.2)	<0.001
	Middle	2151 (4.9)	1965 (4.8)	186 (5.8)	
	Lower	14,107 (31.9)	13,146 (32.1)	961 (30.0)	
	Other	540 (1.2)	287 (0.7)	253 (7.9)	
	Unknown	1045 (2.4)	912 (2.2)	133 (4.1)	
**Age (year)**	<65	14,182 (32.1)	12,913 (31.5)	1269 (39.6)	<0.001
	>64	30,021 (67.9)	28,083 (68.5)	1938 (60.4)	
**Surgical type**	None	20,282 (45.9)	17,981 (43.9)	2301 (71.7)	<0.001
	Limited resection	4007 (9.1)	3933 (9.6)	74 (2.3)	
	Lobectomy	18,681 (42.3)	18,098 (44.1)	583 (18.2)	
	Pneumonectomy	972 (2.2)	744 (1.8)	228 (7.1)	
	Unknown	261 (0.5)	240 (0.6)	21 (0.6)	
**Radiotherapy**	None	28,447 (64.4)	27,034 (65.9)	1,413 (44.1)	<0.001
	Yes	15,442 (34.9)	13,678 (33.4)	764 (55.0)	
	Unknown	314 (0.7)	284 (0.7)	30 (0.9)	
**Chemotherapy**	None	30,828 (69.7)	29,317 (71.5)	1511 (47.1)	<0.001
	Yes	13,375 (30.3)	11,679 (28.5)	1696 (52.9)	
**Marital status**	None	19,729 (44.6)	18,216 (44.4)	1513 (47.2)	0.004
	Married	22,769 (51.5)	21,179 (51.7)	1590 (49.6)	
	Unknown	1705 (3.9)	1601 (3.9)	104 (3.2)	
**Grade**	Well	4570 (10.3)	4405 (10.7)	165 (5.1)	<0.001
	Moderate	13,147 (29.7)	12,373 (30.2)	774 (24.1)	
	Poor	14,215 (32.2)	13,053 (31.8)	1162 (36.2)	
	Undifferentiated	12,271 (27.8)	11,165 (27.3)	1106 (34.6)	
**Tumor size (cm)**	≤3.0	25,321 (57.3)	24,549 (59.9)	772 (24.1)	<0.001
	3.1–4.0	7246 (16.4)	6621 (16.2)	625 (19.5)	
	4.1–5.0	4472 (10.1)	3920 (9.6)	552 (17.2)	
	5.1–7.0	4565 (10.3)	3858 (9.3)	707 (22.0)	
	≥7.1	2599 (5.9)	2048 (5.0)	551 (17.2)	
**N classification**	N0	29,208 (66.1)	27,987 (68.3)	1221 (38.1)	<0.001
	N1	4567 (10.3)	4083 (10.0)	484 (15.1)	
	N2	10,428 (23.6)	8926 (21.7)	1502 (46.8)	
**Histological type**	ADC	21,333 (48.3)	20,435 (49.8)	898 (28.0)	<0.001
	SCC	14,603 (33.0)	12,954 (31.6)	1649 (51.4)	
	Other NSCLCs	3576 (8.1)	3365 (8.2)	211 (6.6)	
	Unknown	4691 (10.6)	4242 (10.4)	449 (14.0)	

ADC: adenocarcinoma, SCC: squamous cell carcinoma, NSCLC: non-small cell lung cancer. * The *p*-value of this variable was calculated by Fisher’s exact test. The p-values of other variables were calculated by the chi-square test.

**Table 2 biomedicines-11-00333-t002:** Multivariable Logistic regression for a risk ratio of lymph node metastasis.

Variables	Multivariable Analysis			Multivariable Analysis		
	RR	95% CI	*p*-Value	RR	95% CI	*p*-Value
Predict lymph Node Metastasis				Predict Lymph Node Metastasis of N2 Station		
Atelectasis/obstructive pneumonitis						
No	1	reference		1	reference	
Yes	2.413	2.230–2.612	<0.001	2.208	2.041–2.390	<0.001
Sex						
Male	1	reference		1	reference	
Female	0.903	0.866–0.942	<0.001	0.928	0.885–0.973	0.002
Race						
Caucasians	1	reference		1	reference	
Other	1.094	1.037–1.155	0.001	1.167	1.100–1.237	<0.001
Unknown	0.801	0.415–1.545	0.508	0.757	0.346–1.656	0.486
Grade						
Well	1	reference		1	reference	
Moderate	2.514	2.273–2.782	<0.001	2.156	1.904–2.441	<0.001
Poor	3.873	3.503–4.281	<0.001	3.528	3.123–3.986	<0.001
Undifferentiated	5.117	4.625–5.662	<0.001	6.092	5.393–6.880	<0.001
Tumor size (cm)						
≤3.0	1	reference		1	reference	
3.1–4.0	1.844	1.741–1.952	<0.001	1.734	1.626–1.848	<0.001
4.1–5.0	2.304	2.152–2.467	<0.001	2.160	2.005–2.326	<0.001
5.1–7.0	2.883	2.693–3.086	<0.001	2.753	2.561–2.959	<0.001
≥7.1	2.978	2.729–3.249	<0.001	2.858	2.611–3.129	<0.001
Histological type						
ADC	1	reference		1	reference	
SCC	0.897	0.854–0.943	<0.001	0.876	0.829–0.926	<0.001
Other NSCLCs	0.871	0.803–0.944	0.001	0.779	0.710–0.854	<0.001
Unknown	1.155	1.076–1.240	<0.001	1.206	1.119–1.299	<0.001
Age (year)						
<65	1	reference		1	reference	
>64	0.712	0.681–0.745	<0.001	0.760	0.724–0.799	<0.001

RR: risk ratio, CI: confidence interval, ADC: adenocarcinoma, SCC: squamous cell carcinoma, NSCLC: non-small cell lung cancer. Variables with *p*-value <0.05 in univariable analysis were incorporated in multivariable analysis. The Logistic regression’s method was Enter selection.

**Table 3 biomedicines-11-00333-t003:** Multivariable Cox regression for cancer-specific mortality in patients with N0 classification after surgery.

Variables	HR	95% CI	*p*-Value
Radiotherapy			
No	1	reference	
Yes	2.085	1.856–2.343	<0.001
Unknown	1.442	0.907–2.293	0.121
Chemotherapy			
No	1	reference	
Yes	1.309	1.199–1.429	<0.001
Year at diagnosis			0.000
2004–2007	1	reference	
2008–2011	0.801	0.752–0.853	<0.001
2012–2015	0.954	0.885–1.028	0.215
Marital status			0.000
Unmarried	1	reference	
Married	0.876	0.828–0.927	<0.001
Unknown	0.739	0.628–0.871	<0.001
Surgical type			
Limited resection	1	reference	
Lobectomy	0.830	0.773–0.892	<0.001
Pneumonectomy	1.179	0.992–1.401	0.061
Lymph nodes			0.000
<5	1	reference	
5–10	0.815	0.758–0.876	<0.001
11–16	0.826	0.757–0.902	<0.001
>16	0.769	0.693–0.853	<0.001
Unknown	0.850	0.766–0.942	0.002
Risk score			
Low score	1	reference	
High score	1.458	1.363–1.559	<0.001

HR: hazard ratio, CI: confidence interval. Variables with *p*-value < 0.05 in univariable analysis were incorporated in multivariable analysis. The risk score was generated from variables (sex, age, grade, tumor size, race, histological type, and atelectasis/obstructive pneumonitis). Therefore, the abovementioned 8 factors did not perform the Cox regression analysis. The Cox regression’s method was Enter selection.

## Data Availability

The researchers interested in this study could contact the corresponding author for requiring the data.

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
