# Peer review of "A Nomogram Based on Atelectasis/Obstructive Pneumonitis Could Predict the Metastasis of Lymph Nodes and Postoperative Survival of Pathological N0 Classification in Non-small Cell Lung Cancer Patients"

_biomedicines, 2023, doi:10.3390/biomedicines11020333_

Round 1

Reviewer 1 Report

Well written article but the content of scientific interest needs to be improved. Some doubts arise:

1) why were only patients with pneumonia and/or atelectasis analyzed in the study group? For the purposes of the study, do these data appear as independent variables regarding the influence on the prognosis? Please explain this part better.

2) In line 207 it speaks of the influence on the prognosis not only of pneumonia and/or atelectasis but also of other factors. The topic seems a bit confused in the explanation.

3) Furthermore, what is the purpose of the work? Can the surgeon decide to submit a patient to neoadjuvant only on the basis of a nomogram?

4) Last correction: in line 219 the adverb "several" is repeated twice

Author Response

Thanks for your timely and professional review.

  1. In this study, we mainly explored the impact of atelectasis/obstructive pneumonitis on lymph-node metastasis. However, your comments were constructive. Therefore, we added the survival analysis of atelectasis/obstructive pneumonitis in the revised manuscript. Thank you again.
  2. Thanks for your reminder. We have added the related analysis of atelectasis/obstructive pneumonitis. You can see Figure 5.
  3. We are grateful for the important comment. There are two main problems in the clinical evaluation of NSCLC patients with atelectasis /obstructive pneumonia. One is that when lung cancer is complicated with atelectasis /obstructive pneumonia, it is difficult to determine the specific area of the tumor through CT and other imaging. Second, it is difficult to determine the spread of tumors in lung tissue, especially when mediastinal lymph nodes overlap with atelectasis. Therefore, for NSCLC patients with atelectasis /obstructive pneumonia, there is an urgent need for a good model to help clinicians better evaluate the patient's condition and formulate treatment plans. The model in this study provides a reference for clinical practice and other researchers. In subsequent studies, researchers can add more meaningful variables based on this model. The decision on preoperative neoadjuvant treatment still depends on the results of the lymph node biopsy. Thank you again for your professional comments.
  4. We are sorry for the mistake. We have corrected it in the revised manuscript. We have also gone through the revised manuscript carefully to make sure all errors have been corrected properly.

Reviewer 2 Report

The authors demonstrated the impact of atelectasis/obstructive pneumonitis on the prediction of the metastasis of lymph nodes. I would like to make some comments on this manuscript

Major

The purpose of the author's study is not clear. He states that the factors to be included in the nomogram are limited to those that can be evaluated preoperatively, but why not include imaging evaluation? It would be more accurate if the size of each lymph node and the SUV value from the PET scan were included. On the other hand, why are the histology results, including the grade of malignancy, included? Some histological examination has been performed, then it can be confirmed by bronchoscopy whether the cause of obstructive pneumonia is bronchial compression by metastatic lymph nodes or not, so there is no need for a nomogram in the first place. 

It appears that the authors are merely playing around with the data of the SEER database.

AUC 0.7 is rather poor to detect lymph node metastasis from this nomogram. Since the authors are affiliated with one of the largest medical institutions in China, I believe their own database would be more reliable in showing how valid this nomogram is as an evaluation cohort.

Minor

The table is very difficult to read. The overall format should be changed.

Author Response

Thanks for the constructive and professional comments.

1. Because the data we analyzed is from the SEER database, the results of clinical examination data such as PET scans are not included in the database, which is also our regret. We will carry out relevant retrospective studies as soon as possible in the future to incorporate the results of the PET scan.

There are a number of studies showing the benefits of adaptive radiotherapy in NSCLC patients with atelectasis /obstructive pneumonia[1, 2], but few discuss whether atelectasis /obstructive pneumonia helps physicians judge whether a patient requires neoadjuvant chemotherapy. Our study is meant to explore the potential relationship between atelectasis /obstructive pneumonia and lymph node metastasis in patients, to provide recommendations for clinicians evaluating the lymph node situation, and to develop treatment regimens for patients with NSCLC complicated by atelectasis /obstructive pneumonia.

2. Thanks for the suggestion. We will conduct a retrospective study in our hospital as soon as possible to collect enough data to verify the significance of atelectasis in the overall evaluation and treatment decision of NSCLC patients. However, we need to trace the CT and pathological results of a large number of patients and follow up with these patients. Therefore, we first verified our idea with the SEER database. Although the SEER database has defects, the analysis results still have certain adaptability and reference value. Thank you for your appreciation and trust in our hospital. We will actively carry out follow-up research.

3. We have modified the table. Thank you.

Reference:

  1. Wang, B., D.Q. Wang, M.S. Lin, S.P. Lu, J. Zhang, L. Chen, Q.W. Li, Z.K. Cheng, F.J. Liu, J.Y. Guo, H. Liu, B. Qiu, Accumulation of the delivered dose based on cone-beam CT and deformable image registration for non-small cell lung cancer treated with hypofractionated radiotherapy, BMC Cancer 2020, 20(1) 1112.
  2. Møller, D.S., A.A. Khalil, M.M. Knap, L. Hoffmann, Adaptive radiotherapy of lung cancer patients with pleural effusion or atelectasis, Radiother Oncol 2014, 110(3) 517-22.

Reviewer 3 Report

To the authors:

In the present manuscript, Liu et al have analyzed the risk factors for lymph node metastases in lung cancer. The paper is interesting and worth for publication. However, some important points should be addressed before publication can be recommended from my side

-          One recently published risk factor for lymph node metastases and cancer specific outcome is missing completely, namely the lymph vessel invasion. Do you have the chance to include this important factor to your model? (compare Biesinger et al Cancers 2022, Song et al  J Thorac Dis 2019)

-          Can you comment on the proportion of minimal invasive surgery vs thoracotomy in the 42.3% of lobectomy patients and the limited resection group?

-          Was limited resection the same as anatomic segmentectomy or was it wedge resection or both? Interestingly, limited resection showed the worse outcome compared to lobectomy, please also comment on this important issue, since this is currently also a matter of debate (compare Stamatis et al Lung Cancer 2022 and Suzuki et al J Thorac Cardiovasc Surg 2019)

-          Was there a difference between number of resected lymph nodes between the lobectomy and limited resection group?

-          I would be interested in the prognostic value of atelectasis as standalone factor. Could you provide a KM graph that illustrates the prognostic role of atelectasis since you also mention that it was a staging  factor?

-          You state in line 98 that type of surgery, radiotherapy and chemotherapy influenced the occurrence of atelectasis, I guess that you mean, the factors were associated with each other, since treatment is post hoc

-          Was the study maybe overpowered? Some results like Marital Status vs. Atelectasis might argue for this. What could be the reason that Marital Status as factor influences the presence of the factor Atelectasis? Should be discussed

-          Please correct minor typos like line 200 … play negative prognostics for a prognosis for patients with NSCLC

-          Line 219 the same

Author Response

Thanks for your timely and professional review.

  1. Thanks for the comment. As the data we analyzed is from the SEER database, the lymph vessel invasion was not included in the database, which is also our regret. In addition, the confirmation of the lymph vessel invasion needs the histomorphological evaluation alone or with additional immunohistochemical evaluation. It’s difficult to confirm the lymph vessel invasion before surgery. However, our study mainly constructed a predictive model before surgery. Your suggestions are very instructive for our future work. We will carry out relevant retrospective studies as soon as possible in the future to incorporate the results of the histology.
  2. Sorry, we can't answer this question. Your point is very professional, and we appreciate it very much. However, the SEER database lacks relevant information about open surgery and minimally invasive surgery, so we cannot comment. Sorry again.
  3. Your comments are very professional. The limited section includes the wedge section and the segment section. In this study, we mainly classified the surgical methods according to whether the resection scope includes a lung lobe, and also for the convenience of statistical analysis. In this study, the prognosis of limited protection is indeed worse than that of lobectomy. We believe that this phenomenon is in line with clinical practice. In particular, we have not divided patients into groups. For tumors smaller than 2cm, we will consider limited restoration, as is the case in the following document[1]. However, the selection of patients with limited protection still needs to be confirmed by prospective research.

  4. We performed the student’s t-test to compare the number of resected lymph nodes between the lobectomy and limited resection groups. We found that the mean lymph nodes were 3.42 and 9.49 in the limited resection and lobectomy groups (P <0.001). The inadequate lymph-node resection might lead to unclear N staging. Therefore, we could see that patients with much-resected lymph nodes have better survival (Table 2). Because those patients had reliable N staging.

  5. According to your comments, we have added Figure 5.

  6. Thanks for the comment. This part of our study was to examine the influence of various factors on cancer-specific mortality, so we established this multivariate Cox regression analysis to calculate cancer-specific mortality in patients with N0 classification after surgery. We have modified it.
  7. The study was retrospective. Therefore, the selection bias was inevitable. The distribution of marital status was unbalanced in the grouping of atelectasis/obstructive pneumonitis. We have added it to the discussion. Thank you.

  8. Thanks for the comment. We have corrected it in the revised manuscript.
  9. We are sorry for the mistake. We have corrected it in the revised manuscript. We have also gone through the revised manuscript carefully to make sure all errors have been corrected properly.

Reference:

1. Saji, H., M. Okada, M. Tsuboi, R. Nakajima, K. Suzuki, K. Aokage, T. Aoki, J. Okami, I. Yoshino, H. Ito, N. Okumura, M. Yamaguchi, N. Ikeda, M. Wakabayashi, K. Nakamura, H. Fukuda, S. Nakamura, T. Mitsudomi, S.I. Watanabe, H. Asamura, Segmentectomy versus lobectomy in small-sized peripheral non-small-cell lung cancer (JCOG0802/WJOG4607L): a multicentre, open-label, phase 3, randomised, controlled, non-inferiority trial, Lancet 2022, 399(10335) 1607-1617.

Round 2

Reviewer 2 Report

No comments.